# What makes a temperate phage an effective bacterial weapon?

M. J. N. Thomas,[1] M. A. Brockhurst,[1] K. Z. Coyte[1]

**ABSTRACT** Temperate bacteriophages (phages) are common features of bacterial genomes and can act as self-amplifying biological weapons, killing susceptible competitors and thus increasing the fitness of their bacterial hosts (lysogens). Despite their prevalence, however, the key characteristics of an effective temperate phage weapon remain unclear. Here, we use systematic mathematical analyses coupled with experimental tests to understand what makes an effective temperate phage weapon. We find that effectiveness is controlled by phage life history traits—in particular, the probability of lysis and induction rate—but that the optimal combination of traits varies with the initial frequency of a lysogen within a population. As a consequence, certain phage weapons can be detrimental when their hosts are rare yet beneficial when their hosts are common, while subtle changes in individual life history traits can completely reverse the impact of an individual phage weapon on lysogen fitness. We confirm key predictions of our model experimentally, using temperate phages isolated from the clinically relevant Liverpool epidemic strain of *Pseudomonas aeruginosa*. Through these experiments, we further demonstrate that nutrient availability can also play a critical role in driving frequency-dependent patterns in phage-mediated competition. Together, these findings highlight the complex and context-dependent nature of temperate phage weapons and the importance of both ecological and evolutionary processes in shaping microbial community dynamics more broadly.

**IMPORTANCE** Temperate bacteriophages—viruses that integrate within bacterial DNA—are incredibly common within bacterial genomes and can act as powerful self-amplifying weapons. Bacterial hosts that carry temperate bacteriophages can thus gain a fitness advantage within a given niche by killing competitors. But what makes an effective phage weapon? Here, we first use a simple mathematical model to explore the factors determining bacteriophage weapon utility. Our models suggest that bacteriophage weapons are nuanced and context-dependent; an individual bacteriophage may be beneficial or costly depending upon tiny changes to how it behaves or the bacterial community it inhabits. We then confirm these mathematical predictions experimentally, using phages isolated from cystic fibrosis patients. But, in doing so, we also find that another factor—nutrient availability—plays a key role in shaping bacteriophage-mediated competition. Together, our results provide new insights into how temperate bacteriophages modulate bacterial communities.

**KEYWORDS** bacteriophages, microbial ecology, mathematical modeling, temperate phage, bacterial weapons, microbial evolution

Temperate bacteriophages (phages) provide their bacterial hosts (lysogens) with resistance to genetically similar phages through superinfection exclusion (1–3), enabling their use as self-amplifying biological weapons by lysogens against susceptible competitors (4–7). Consequently, lysogens are predicted to be better than non-lysogens at invading (or defending) ecological niches (6). Lysogens typically produce temperate phage virions at a low rate through spontaneous induction of the lytic cycle in a

Address correspondence to K. Z. Coyte, katharine.coyte@manchester.ac.uk.

The authors declare no conflict of interest.

See the funding table on p. 16.

subpopulation, releasing phage virions into the environment (3, 8, 9). These virions can infect nearby susceptible bacteria and replicate through the lytic cycle, killing these susceptible cells and thus enabling the lysogen to outcompete its susceptible neighbors. The utility of temperate phage weapons has been shown both in the lab and within animal infection models (4, 5). Indeed, carrying multiple temperate phages is thought to be an important determinant of the fitness of the polylysogenic Liverpool epidemic strain (LES) of *Pseudomonas aeruginosa* within cystic fibrosis patients (10). Virions are actively produced in LES human lung infections, and lysogens have been shown to be more competitive in experimental rat lung infections (8). As such, through their effects mediating bacterial competition, temperate phages contribute to controlling the dynamics of both host-associated and environmental microbial communities (11).

While there is good evidence that lysogens can be more competitive, it is less clear how the various life-history traits of temperate phages contribute to this fitness benefit. Some work has begun to address these questions through mathematical modeling. For example, de Sousa and colleagues (7) coupled individual-based models with random forest analyses to explore how variability in traits such as the probability of lysis (the ratio of lytic to lysogenic infections), the phage induction rate (how rapidly new phages are produced), or the absorption rate (how rapidly free phages bind to susceptible cells) contributed to variability in the number of lysogenized resident bacteria. This work found that variability in key traits such as the probability of lysis and burst size (how many phage virions are produced upon induction) tended to drive the greatest variability in lysogen fitness. However, whether and how much each individual life history trait altered the utility of a given temperate phage as a weapon remained unclear. Similarly, ordinary differential equation models demonstrated that the initial frequency of a lysogen is also an important factor in determining whether it can invade a resident population, with sufficient susceptible bacteria required within a population to generate exponential amplification of the phage (6). Together, these mathematical results suggest both invasion conditions and phage characteristics are likely to play a critical role during lysogen invasion. Yet, despite this, we still lack any systematic study of what makes a useful temperate phage weapon.

Here, we combine mathematical modeling and *in vitro* experiments to explore comprehensively how different life-history traits and ecological contexts intersect to determine the utility of a temperate phage as a weapon. Our mathematical modeling confirms previous observations that different life history traits have very different effects on phage utility. However, we now find these impacts are highly dependent upon the frequency of the lysogen in the population. We find that certain phages may be beneficial under some circumstances yet detrimental in others and that the "best" phage varies depending on initial population conditions. We confirm these mathematical predictions experimentally, using clinically relevant bacterial strains to confirm that the probability of lysis plays a key role in determining phage utility, but in a manner that is strongly frequency dependent. Surprisingly though, these experiments reveal that an additional, often overlooked environmental factor—variation in nutrient availability—also has a critical role in modulating the impact of a phage upon lysogen dynamics. Together, our results reveal the importance of the interplay between the properties of individual phages, their hosts, and their broader environments in determining the effect of temperate phages within microbial communities.

## RESULTS

### Mathematical modeling allows us to explore the determinants of phage utility

To understand how different phage life history traits and environmental factors impact the effectiveness of a temperate phage as a weapon, we adapted a previously published mathematical model of a single lysogen growing within a population of otherwise susceptible bacteria (Fig. 1; Table 1, Materials and Methods) (6). Here, the underlying growth of each strain within the community is determined by its own intrinsic growth

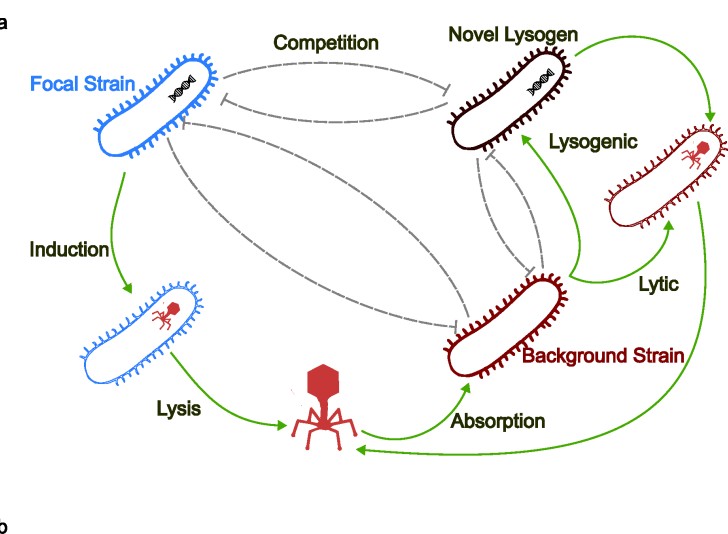

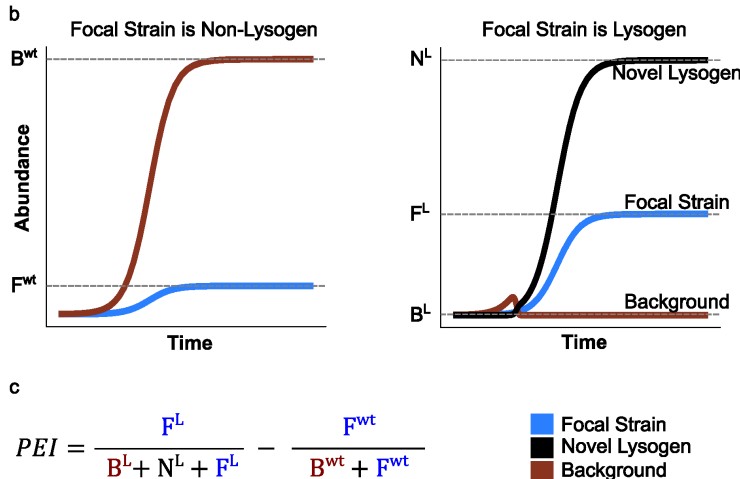

$$PEI = \frac{F^L}{B^L + N^L + F^L} - \frac{F^{wt}}{B^{wt} + F^{wt}}$$

Focal Strain
Novel Lysogen
Background

**FIG 1** Mathematical modeling captures the dynamics of phage-mediated competition. (a) Schematic illustrating our mathematical model. Here, we focus on the dynamics of two competing strains, an initially susceptible "Background" strain (dark red) and a "Focal" strain (dark blue), which may or may not carry a temperate phage weapon. (b) Example bacterial population dynamics from simulations when the Focal strain does (right—L) or does not (left—WT) carry a temperate phage weapon. Dashed lines indicate final population abundances, which are in turn used to calculate the phage effectivity index (PEI). (c) Formula for the PEI is defined as the difference in the final frequency of the Focal strain with vs without a temperate phage. Here, F indicates the final abundance of the focal strain, B indicates the background strain, and N indicates the final abundance of the novel lysogen formed when the phage integrates into the background strain. Superscripts indicate simulations in which the Focal strain does (L) or does not (WT) harbor a temperate phage.

rate, $r$, and any interactions with self and other community members (encoded by the matrix $\mu$), each of which is assumed to be competitive and reciprocal. We assume that each lysogenic cell induces at a rate $\kappa$, forming a latent infected cell, which lyses at a rate $\lambda$, producing free phage virions. Free phage virions are produced at a burst size, $\beta$ and are degraded (naturally lost) at a rate $\delta$. Free phage virions can then be absorbed by susceptible cells at a rate $\alpha$. With a probability $\pi$, these newly infected cells lyse, while with a probability $1 - \pi$, they form new lysogens. This yields the following system of ordinary differential equations, which can be numerically solved to simulate the dynamics of any given set of lysogenic and non-lysogenic strains over time:

$$\text{Focal strain} = \frac{dF}{dt} = F(r + F\mu - \kappa)$$

Background strain $= \dfrac{dB}{dt} = B(r + B\mu - a\pi P - a(1 - \pi)P)$

Novel lysogen $= \dfrac{dN}{dt} = Ba(1 - \pi)P + N(r + N\mu - \kappa)$

Bacteriophage $= \dfrac{dP}{dt} = \beta\lambda\tilde{B} + \beta\lambda\tilde{F} - aPB - \delta P$

Induced/infected background strain $= \dfrac{d\tilde{B}}{dt} = \kappa N + a\pi BP - \lambda\tilde{B}$

Induced focal strain $= \dfrac{d\tilde{F}}{dt} = \kappa F - \lambda\tilde{F}$

We used this model to explore the simple scenario whereby two strains compete against one another within a niche. In our model, we assumed that one strain (the Background [B]) was always non-lysogenic, while the other strain (the Focal [F]) could either be non-lysogenic or be a lysogen carrying a temperate phage weapon. We performed these competitions under a series of different starting conditions, varying the ratio of the Focal to Background strains (F:B) to capture scenarios ranging from the initial invasion of the Focal strain into a new niche (1:99 F:B), through an evenly pitched battle (50:50 F:B), to the Focal strain defending its niche against an invading Background strain (99:1 F:B).

Comparing the outcome of these competitions when the Focal strain was lysogenic to when it was non-lysogenic thus enabled us to quantify the relative advantage (or disadvantage) to a bacterium possessing a given temperate phage, which we term the phage effectivity index (PEI) (Fig. 1c).

We focused on this measure as it allowed us to directly capture the absolute advantage of a focal strain carrying a phage weapon (compared to a phage free non-lysogen) regardless of context. This metric has an intrinsic frequency dependence: because there is an upper limit on frequency within the population, strains that are initially common can only increase in frequency by a small amount, regardless of the effectivity of their phage weapon. Therefore, for completeness, we also assessed phage effectiveness under two further metrics. First, we calculated PEI as the relative fitness ($v$, Fig. S1 and S2) of the lysogen compared to the background strain during each competition. Second, we calculated PEI as the natural log of the focal strain's fold change in abundance within the population, giving an estimate of the average growth rate of the focal strain during each competition (Fig. S3).

## Life history traits and invasion frequency combine to shape the impact of phage weapons

Systematically varying phage life history traits revealed that, as expected, each trait differed in both the direction and the magnitude of its impact on PEI. In line with previous results (7), we found that small changes in the probability of lysis ($\pi$) and the

**TABLE 1** Parameters used for model simulations

| Parameter | Symbol | Experimental parameters | | Variable parameters | | |
|---|---|---|---|---|---|---|
| | | Phi2 | Phi4 | Fixed | Min | Max |
| Probability of lysis | $\pi$ | 0.91 | 0.72 | 0.5 | 0.01 | 0.99 |
| Induction rate | $\kappa$ | 0.0001 | 0.0001 | 0.0001 | 0.00001 | 0.9 |
| Lysis rate | $\lambda$ | 0.505 | 0.505 | 0.505 | 0.01 | 1 |
| Absorption rate | $\alpha$ | $1 \times 10^{-8}$ | $1 \times 10^{-8}$ | $1 \times 10^{-8}$ | $1 \times 10^{-10}$ | $1 \times 10^{-6}$ |
| Degradation rate | $\delta$ | 0.01 | 0.01 | 0.05 | 0.00001 | 0.9 |
| Burst size | $\beta$ | 100 | 100 | 100 | 10 | 1,010 |
| Intrinsic growth rate | $r$ | 0.9 | 0.9 | NA | NA | NA |
| Interaction matrix | $\mu$ | −4E−10 | −4E−10 | NA | NA | NA |

*a*"Experimental parameters" indicate the values used to simulate Phi2 and Phi4 for comparison against experimental data. "Variable parameters" indicate the ranges used when exploring parameter space, with the min and max columns showing the range of parameters explored, the fixed column showing the values chosen when that parameter was not varied. Intrinsic growth rates and interaction terms were never varied; thus, "NA" indicates not applicable.

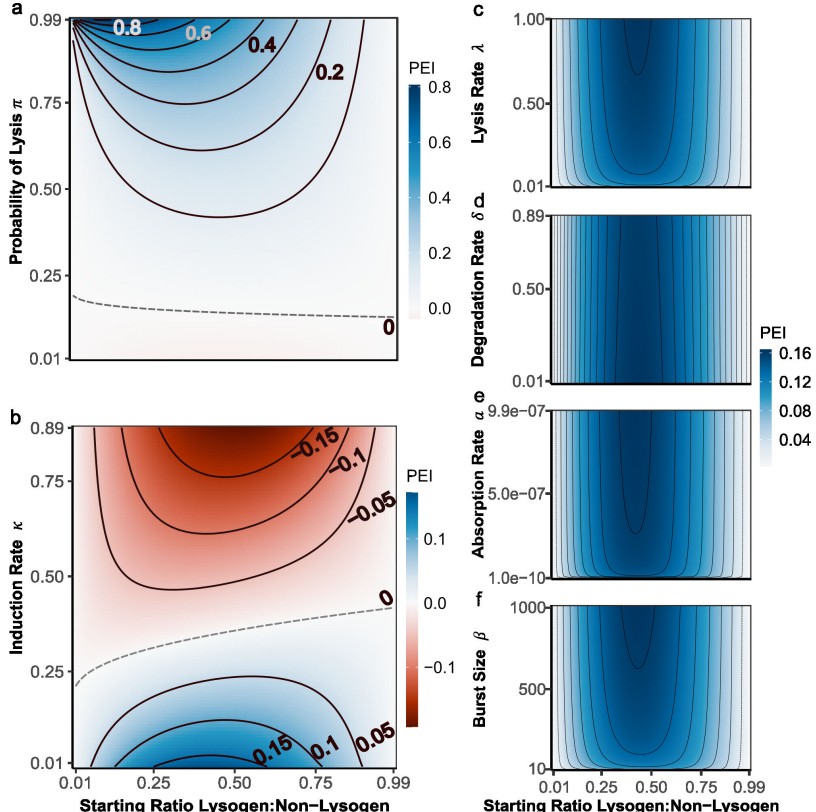

**FIG 2** Phage life history traits and lysogen frequency jointly determine the effectiveness of a temperate phage as a bacterial weapon (PEI). Each heatmap illustrates the impact of varying a given life history trait (y-axis) and invasion frequency (x-axis) upon the effectiveness of a temperate phage weapon. Blue indicates that the phage weapon is beneficial, while red indicates that it is costly. Contours (black lines) show different PEI values, with dashed lines corresponding to PEI = 0. Subpanels correspond to varying (a) probability of lysis, (b) induction rate, (c) lysis rate, (d) degradation rate, (e) absorption rate, and (f) burst size. Parameters for each simulation are given in Table 1.

induction rate ($\kappa$) both led to large changes in the effect of a given phage, yet equivalent changes to traits such as the lysis rate ($\lambda$) or degradation rate ($\delta$) had virtually no impact upon PEI (Fig. 2). Expanding on this previous work, our analysis now also demonstrated that increasing the probability of lysis consistently increased the effectiveness of a phage as a weapon via reducing the rate at which novel resident lysogens were formed (Fig. 2a). In contrast, increasing induction rate typically substantially decreased PEI—to the extent that lysogens with a high induction rate were consistently less fit than the non-lysogenic control (negative PEI, Fig. 2b). That is, our analysis demonstrated that while temperate phages are beneficial on average, harboring a lysogenic phage with a high induction rate can be worse than harboring no phage at all.

Our analysis also revealed that the effect of harboring a phage on lysogen competitiveness was strongly dependent upon their initial frequency within the population. In most cases, phage provided the greatest advantage when lysogens were at intermediate frequencies. In contrast, when lysogens were initially present at either very low or very high frequencies, we observed very little impact of harboring a phage upon the success of a lysogen within a population (PEI ≈ 0). The exact shape of this frequency dependence varied somewhat depending on the measure of phage effectivity used. For example, lysogen relative fitness still peaked at intermediate lysogen frequencies but reduced the magnitude of this effect (Fig. S1 and S2). In contrast, the fold change in focal strain frequency often suggested that phage weapons are most effective when the focal strains

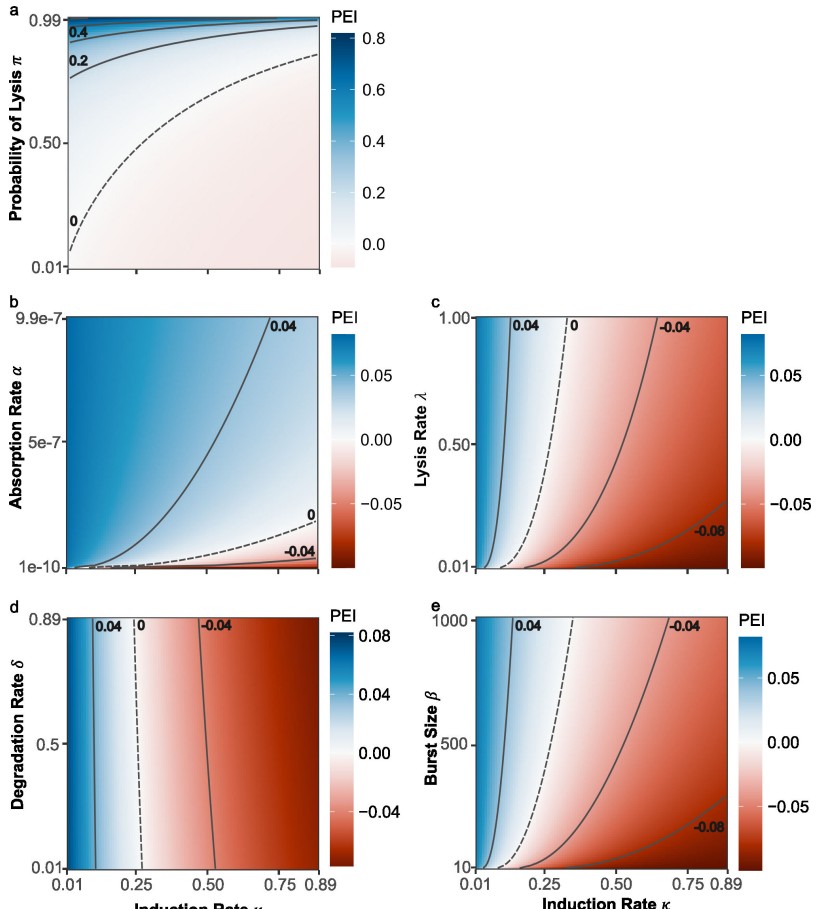

**FIG 3** Interplay between different life history traits allows some traits to be compensatory. Here, each heatmap illustrates how varying different life history traits (*y*-axis) against different induction rates (*x*-axis) changes the efficacy of different phages as weapons. Blue shows that the weapon is highly effective, whereas red indicates the weapon is costly. Contours (black lines) show different PEI values, with dashed lines corresponding to PEI = 0. Subpanels correspond to (a) probability of lysis, (b) absorption rate, (c) lysis rate, (d) degradation rate, and (e) burst size. All simulations are performed with a Focal:Background starting ratio 0.1:0.9; all other parameters are given in Table 1.

are rare (Fig. S3), consistent with previous observations from Brown and colleagues (6) and likely reflecting that fold changes are over-emphasized at small values.

Importantly, however, regardless of the PEI metric used, we found that the relationship between individual phage life-history traits and phage weapon effectivity was strongly modulated by initial lysogen frequency. For example, while harboring a phage with a high induction rate was always detrimental, this disadvantage was much lower when lysogens were common within the population than when they were rare. This interplay between the characteristics of a lysogen and its environment led to situations whereby a given phage could either be costly or beneficial, depending on the starting frequency of the lysogen within the population. For example, we found that a phage with an induction rate of 0.3 could be costly to a lysogen when its frequency within the population was low, yet beneficial when lysogen frequencies were high (Fig. 2b). Together our analyses suggest that the impact of a temperate phage weapon on its host's fitness is strongly dependent not only on the characteristics of the phage but also the ecological context in which both bacterium and phage exist.

## Life history traits can have compensatory effects

Given the very different impacts of different life history traits upon phage weapon effectiveness, we next sought to investigate how these different traits modulate one another's effects. More specifically, we explored whether other life history traits might be able to mitigate the detrimental effect of high phage induction rates. Regardless of starting frequency, almost all traits were able to mitigate high induction rates; however, some were more effective than others (Fig. 3; Fig. S5). As expected, increasing the probability of lysis ($\pi$) ameliorated the impact of a high induction rate, in some cases switching the PEI of an individual phage from negative to positive (Fig. 3a). More surprisingly, traits such as absorption rate and lysis rate, which had very little effect on PEI when varied in isolation, could also mitigate the negative effects of a high induction rate (Fig. 3b and c). For example, small increases in the absorption rate switched the PEI of high induction rate phages from negative to positive. Notably, the detrimental effect of high induction rates in the focal strain could also be overcome when the novel lysogens possessed even higher induction rates, a phenomenon that has been observed several times in nature (Fig. S4). The only trait that did not have a marked mitigating effect was degradation rate (Fig. 3d). Together, the traits able to mitigate high induction rates appeared to be those that increase the killing of competitors. In other words, our analysis suggested that lysogens can bear the burden of a high induction rate phage, provided this high induction also enables them to rapidly infect and kill competitors and thus clear space within the niche.

## The "optimal" phage varies depending on initial population conditions

Our mathematical model suggested that the overall effectiveness of a temperate phage as a weapon is determined by a complex interplay between different phage life history traits and the initial frequency of its corresponding lysogen within a population. This led us to wonder whether this interplay between different parameters could lead to scenarios whereby the "optimal" phage weapon varied depending upon the ecological context in which a given competition was occurring. To investigate this, we randomly generated 50 different phages, each with a different unique combination of life history traits, then competed the corresponding lysogen of each phage against a non-lysogenic partner at a series of different starting frequencies. At each frequency, we ranked the phage by their PEI, with 1 representing the "most useful" phage (highest PEI) and 50 representing the "least useful" (lowest PEI), such that changes in phage rank with frequency would indicate scenarios where the "optimal" phage weapon changed under different conditions.

We observed multiple PEI rank changes across starting frequencies (Fig. 4a). Some phages steadily dropped in rank with increasing starting frequency—suggesting that these phages conferred a particular advantage over others when rare, but not when common. Other phages showed the opposite pattern, slowly rising through the ranks as lysogen starting frequency increased, while for some phages their PEI rank did not vary with starting frequency. To identify whether there were trends in these PEI rank changes, we repeated our ranking analysis 100 times, creating 100 sets of 100 random phages. We then quantified the number of phages that changed in PEI rank at each starting frequency for each phage set, allowing us to determine a measure of average rank volatility. Here, a rank volatility $\approx 0$ means that the relative utility of different phages does not vary with changes in lysogen starting frequency, whereas a rank volatility $\gg 0$ implies small changes in starting frequency lead to large-scale reordering of the relative utility of different phages as weapons. Our analysis revealed a clear inverse correlation between starting frequency and PEI rank volatility [fitted linear regression model rank volatility (RV) = 62.46–51.65 × (starting ratio), $t(998) = -59.25$, $P < 0.001$, $R^2 = 0.78$]. Small changes in life history traits at low starting frequencies (e.g., from 0.01 to 0.05) were highly likely to alter the relative utility of different phage weapons. Conversely, at high starting frequencies, the relative utility of different phages remained relatively stable.

Altogether, our analysis suggests that different phage weapons are better in different environments, such that subtle differences in individual phage life history traits have substantial impacts on whether and how a lysogen initially invades a population but matter less once lysogens are dominant and defending a niche.

### *In vitro* experiments confirm the predicted impact of the probability of lysis and lysogen starting frequency on phage weapon effectiveness

Our mathematical modeling enabled us to predict PEI as a function of life history traits and initial lysogen frequency. We next sought to test the accuracy of these predictions using a simple *in vitro* experimental system, competing strains of *Pseudomonas aeruginosa* PAO1 with or without each of two different temperate phages (LESB58Φ2 or LESB58Φ4). First, we quantified key life-history parameters for these phages and their corresponding lysogens, revealing that the lysogens only significantly differ in their probability of lysis ($\pi$), with Phi2 having a higher probability of lysis than Phi4 (Fig. 5a through e). We then used these life-history data to qualitatively parameterize our model (Materials and Methods) and then predict how the effectivity of each phage weapon varies under a set of different competition conditions. For completeness, we also used these experimental parameters to repeat our earlier life history trait parameter sweep analysis (Fig. S6).

Our model predicted that Phi2 should consistently display a significantly higher PEI than Phi4, but that the difference in the effectiveness of each phage compared with both a susceptible strain and one another should be greatest at intermediate starting frequencies. Next, we tested these predictions experimentally by competing the lysogens against non-lysogenic PAO1 across a range of starting frequencies. Our experiments showed a good qualitative and quantitative match for each phage (Fig. 5f), with Phi2 showing higher PEI than Phi4, and the PEI for both phages peaking at intermediate starting frequencies (0.1 for Phi2 vs 0.5 for Phi4). These results confirmed our specific predictions regarding the impact of the probability of lysis and initial lysogen frequency on PEI and more generally suggested that our simple model could accurately predict phage utility.

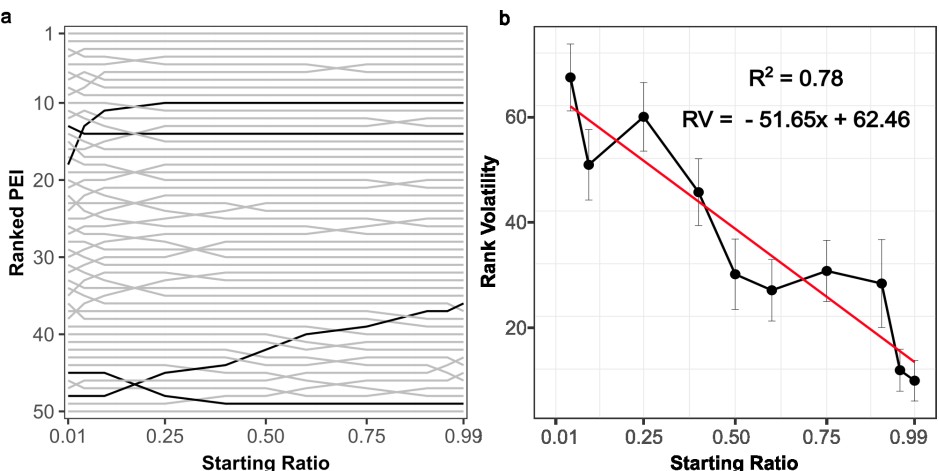

**FIG 4** The utility of different phages changes at different starting ratios. (a) Graph of the ranked utility (*y*-axis) of 50 randomly generated phages at different starting ratios (*x*-axis). Rank 1 corresponds to the most effective phage weapon and rank 50 to the least effective. Crossing of lines indicates that the relative effectiveness of different phage weapons is different at different starting ratios. (b) Graph of the average number of rank changes (rank volatility [*y*-axis]) across starting ratio (*x*-axis) averaged from 100 repetitions of panel a. Red line shows the fitted linear regression model, indicating a negative correlation between the rank volatility and the starting ratio. Parameters for each simulation are given in Table 1.

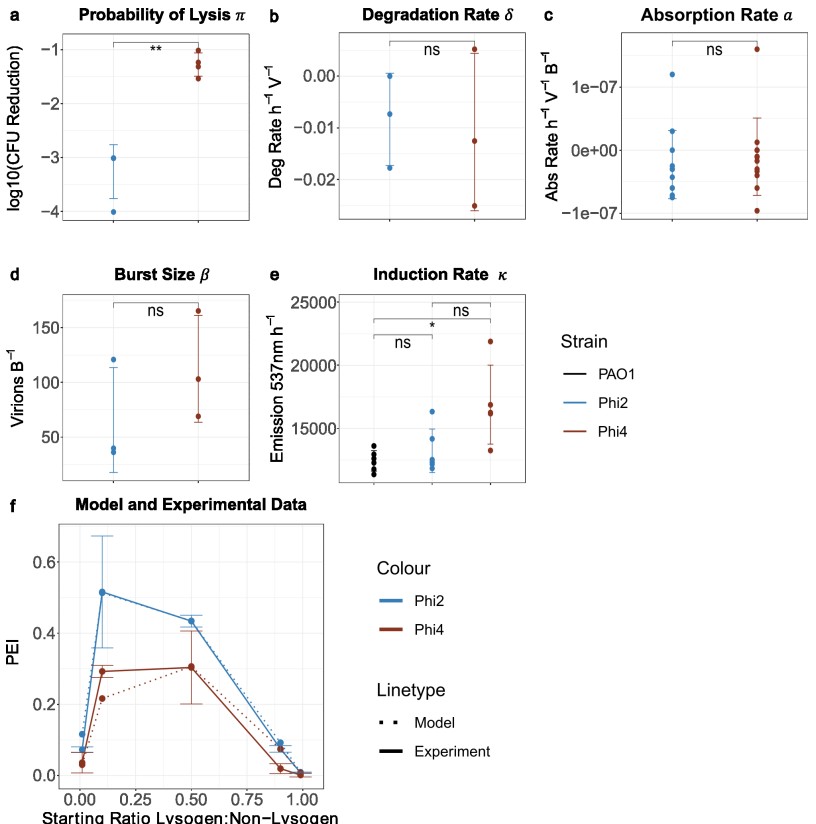

**FIG 5** *In vitro* validation confirms that our mathematical model can qualitatively predict the utility of different phages under different conditions. (a–e) Experimental estimates of each life history trait suggest that Phi2 and Phi4 differ only in their probability of lysis ($\pi$)). Subpanels correspond to (a) probability of lysis (**$P = 0.0017$), (b) degradation rate ($P = 0.8247$, ns), (c) absorption rate ($P = 0.6032$, ns), (d) burst size ($P = 0.3014$, ns), (e) induction rate [$P = 0.34$, ns (PAO1/Phi2), *$P = 0.031$ (PAO1/Phi4), and $P = 0.058$, ns (Phi2/Phi4)], and (f) predicted (dashed lines) and measured (solid lines) phage effectivity indices for Phi2 and Phi4. Predicted and measured PEIs both indicate that Phi2 is more effective as a weapon than Phi4, with this advantage most evident when lysogens are present at intermediate frequencies within the population.

## Microbial dynamics suggest that nutrient availability also shapes lysogen competition dynamics

While our experimental data qualitatively matched our mathematical predictions, we observed an interesting discrepancy between the underlying population dynamics predicted by our models and those we observed experimentally. Specifically, in control experiments, when the GFP-labeled non-lysogen PAO1 focal strain was introduced to a dTomato-labeled PAO1 resident population, the overall bacterial density dynamics (measured by OD$_{600}$ absorbance) followed near-identical classic logistic growth regardless of the focal strain's starting frequency (Fig. 6a). In contrast, when the focal strain was a lysogen, the density dynamics of the overall population varied depending upon the focal strain's starting frequency (Fig. 6b and c, OD ~ strain × ratio, $F_{5,48} = 70.77$, $P < 0.001$). For lysogenic focal strains, as starting frequency reduced, populations achieved lower final densities (Fig. 6d through f) and also showed stronger initial reductions in density early in the growth curve. In our simulations, however, we observed no effect of lysogen starting frequency on final population density (Fig. 6g), suggesting that our model (and thus previous models) is missing a critical component of the ecology of phage-mediated microbial dynamics.

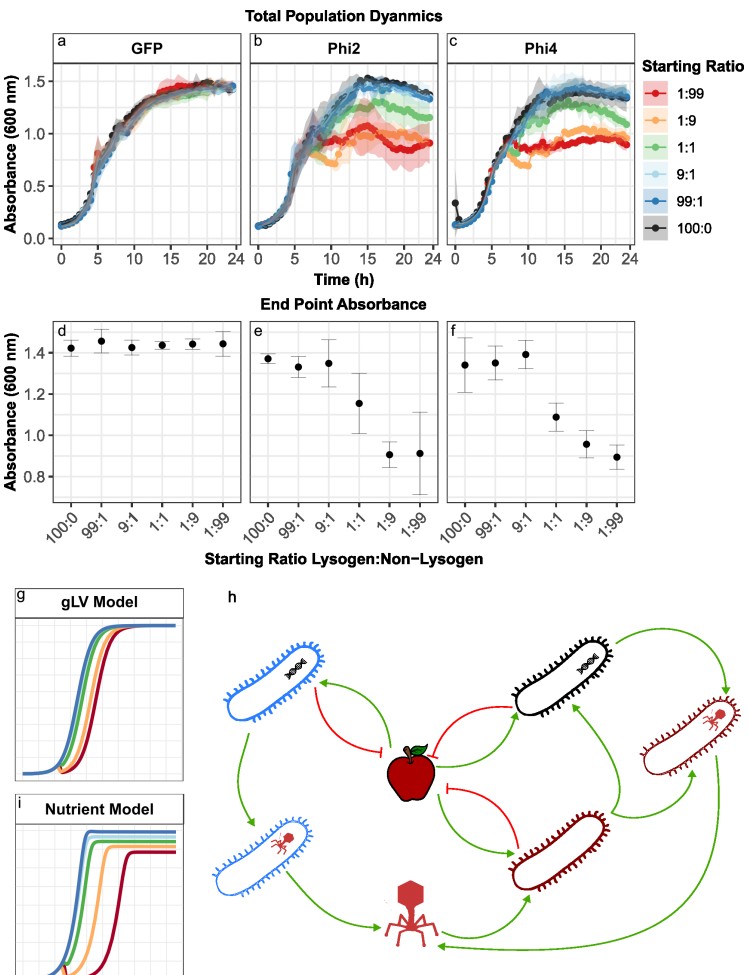

**FIG 6** Growth curves from competition experiments reveal a relationship between initial lysogen frequency and overall population dynamics. (a–c) Bacterial density dynamics (OD$_{600}$, *y*-axis) for different focal strains competing against a background strain over 24 h (*x*-axis). (d–f) Comparison of the final bacterial density of the single infection control (starting ratio 100:0) to the final bacterial density of different competition experiments when the focal strain is non-lysogenic (d), harbors Phi2 (e), or harbors Phi4 (f). The non-lysogenic control shows no reduction in final density across the starting ratios; however, lysogens exhibit a correlation between starting frequency and final bacterial density. That is, when lysogens are initially rare, final bacterial densities tend to be lower (g) Predicted growth dynamics for our original model. (h) Schematic diagram of our model modified to explicitly include nutrient-mediated competition. (i) Predicted population growth dynamics for the nutrient-explicit model.

We hypothesized that this impact on final densities could be driven by the "seasonal" batch culture nature of our experiments, whereby each population is growing in an environment with a fixed starting level of nutrients that will be steadily consumed over time. In contrast, our mathematical model is based upon the classic generalized Lotka-Volterra equations (12, 13), which are known to be best suited to continuous culture conditions such as those of a chemostat but are less well-suited to batch culture environments (14). To explore whether this might be the case, we altered our initial model such that the growth of each strain was now dependent upon the explicit uptake and utilization of a single nutrient within the medium (Fig. 6h). Importantly, our goal here was not to capture any one specific metabolite within the environment, as we could not easily determine precisely which individual (or multiple) metabolites were being utilized within our system. Rather, we aimed to qualitatively capture one potential mechanism that might be controlling overall population dynamics.

Our nutrient-explicit model could qualitatively reproduce each of the characteristic density dynamics observed within our experiments—reproducing the correlation between lysogen starting frequency and final density, and the "dip" in densities observed at approximately 8 h (Fig. 6e). Crucially, this alteration to our model did not alter our predictions regarding phage effectiveness, producing near-identical relationships between life history traits, lysogen starting frequency, and PEI (Fig. S7 and S8). While our new model only captures one potential scenario, these results suggest that when lysogens are initially rare, susceptible cells are able to grow for some time—consuming nutrients within the environment—before they are infected and ultimately killed by the phage. These susceptible cells may therefore act as a nutrient sink, reducing the energy available to both the invading lysogen and newly lysogenized surviving cells, thus limiting the final density the total population can reach. In contrast, when lysogens dominate the population and/or susceptible residents are not killed by phage infection (as in the case of the control competitions against a non-lysogen strain), there is less potential for phage-mediated killing to remove consumed nutrients from the environmental nutrient pool, enabling higher final cell densities. Further experimental evidence will be required to fully assess the validity of our new model. Nonetheless, our results further support our conclusion that ecological context (both in terms of starting frequencies and factors such as nutrient availability) can play a critical role in determining the utility of a temperate phage weapon and its subsequent impact on overall community dynamics.

## DISCUSSION

By some estimates, most bacteria are lysogens (15). Although prophage regions can degrade through evolution (16), deactivating the phage and preventing lysis, many bacterial genomes retain active prophage(s) for long periods of time despite the risk of imminent death (8). A case in point is the LES, which even after decades in the lung of CF patients maintains prophage capable of inducing lysis and releasing phage virions (8). The only plausible explanation for the long-term maintenance of active lysogeny by bacteria is that the temperate phages they harbor provide them with a fitness benefit (17, 18). One potential benefit is that the temperate phages act as self-amplifying weapons that give lysogens a competitive edge by killing susceptible competitors. As such, temperate phages boost lysogen fitness both during the initial colonization and the subsequent defense of ecological niches, with evidence for this effect accumulating from a diverse range of systems and environments (5, 6, 8, 11). Yet, despite their ecological importance, precisely what makes a useful temperate phage weapon has remained unclear. Here, we combined simple mathematical models and experiments to disentangle how both phage life history traits and ecological context shape the effectiveness of temperate phages as biological weapons.

Consistent with previous work, our results suggest that the optimal phage weapons are typically those with a low induction rate but a high probability of lysing competitor cells (7). That is, the best weapons are those that kill the lysogen as little as possible while rapidly killing competitors and minimizing the formation of new lysogens resistant to the weapon. However, our results also reveal that phage effectiveness is a nuanced and multi-dimensional property. Detrimental phage weapons (such as those with high induction rates) may be rendered beneficial through small changes in other life history traits (such as absorption rate), creating the potential for intricate evolutionary trajectories. More fundamentally, we find that the effectiveness of a phage weapon is not universal, but rather is contingent upon ecological context. The same phage weapon may be beneficial or costly depending upon the starting frequency of its lysogen within a population, and the relative ranking of different phage weapons switches with different starting conditions. It is worth noting that in our simulations, the variation in phage weapon effectiveness between different phages, and between different conditions, is relatively small. Though even subtle differences in an organism's fitness could have substantial impacts upon its long-term dynamics, to fully understand

the causes and consequences of phage weapon diversity the next step will be to directly compete lysogens with different strategies against one another within an evolutionary modeling framework. Nonetheless, our work suggests that these context-dependent determinants of a phage weapon's effectiveness may be an important contributor to the large diversity in temperate phage life history traits observed in nature.

An unexpected finding was the key role of nutrients—and more specifically, dynamic changes in nutrient availability over time—in modulating the interplay between phage life history traits, initial conditions, and overall population dynamics. Comparing our simulations and experiments suggests that, in the presence of lysogens, the consumption of nutrients by non-lysogenic competitors that are subsequently lysed effectively removes energy from the system and thus generates a correlation between initial lysogen frequency and final population densities. While this impact on population density did not appear to affect the overall utility of a given phage weapon in our simple two strain competitions, we suspect these dynamics could become much more important in a community context. Most microbes inhabit complex multispecies communities, competing and cooperating with numerous other taxa; as such, if lysogen invasion reduces the abundance of one taxon, this disturbance may have a knock-on effect upon many other taxa (19). Understanding how these phage-nutrient dynamics play out within a broader community or meta-community model is therefore an important future avenue for research.

Here we have focused on how the characteristics of a given temperate phage impact the fitness of its bacterial host. However, an equally important question is how these life history traits impact the spread and abundance of the phage itself. While a formal analysis of optimal phage strategies is beyond the scope of this paper, we can infer a component of phage fitness by simply tracking the number of free phage virions produced across each of our simulations, which suggests phage fitness is similarly dependent upon ecological context. Notably, whereas bacteria benefit from very low induction rates, phage fitness, at least in some cases, is maximized at higher induction rates (Fig. S10). This divergence is particularly interesting as it suggests the potential for conflicts of interest between temperate phages and their bacterial hosts over what is the optimal life history. To fully understand how conflicting phage and bacterial fitness interests shape evolutionary trajectories will require a more formal coevolutionary analysis; however, our results suggest that such conflicts may be yet another important factor driving the phage diversity observed in nature (20).

To explore systematically the determinants of phage weapon effectiveness, here we have used relatively simple ecological models. The advantage of these simple models is that we can screen large numbers of temperate phage life history strategies in high throughput using *in silico* experiments. As a consequence, we have omitted a number of features such as density-dependent phage induction and *de novo* phage resistance that are known to occur within natural systems (21–24). We also focus here only on the initial ecological dynamics of invasion, but the impacts of phage life history on PEI are also likely to vary over shorter or longer timescales (Fig. S9). Understanding how these additional layers of complexity further modulate the effectiveness of phage weapons is therefore an important open question. However, it is notable that despite their simplicity, our models accurately predicted our experimental results. This suggests that our overarching predictions—namely that the effectiveness of a given phage weapon depends not only upon its individual life history traits but also the ecological context in which its host resides—are likely to hold under a wide range of different biological conditions. More generally, our results underscore the power of simple mathematical models to understand the drivers of microbial dynamics.

## MATERIALS AND METHODS

### Initial model analysis

We initially explored bacterial and phage dynamics using a simple mathematical model first developed by Brown et al. (6). This approach uses a system of ordinary differential equations to capture the dynamics of a focal strain (F), a background strain (B), and a novel lysogen (N) formed when the bacteriophage (P) infects the background strain through the lysogenic cycle. The model also contains explicit compartments for the focal strain upon phage induction ($\widetilde{F}$) and the background strain following either phage induction or infection via the lytic cycle ($\widetilde{B}$). Notably, here we rewrite Brown et al.'s original equations to take the form of the generalized Lotka-Volterra equations, so as to allow easy comparison with previous microbial community models. However, this alternative functional form is mathematically equivalent to the original Brown et al.'s approach of logistic growth with a shared carrying capacity. Bacterial and phage life history trait parameters are defined in the text and in Table 1.

For each simulation, we initialized the system with a low-density bacterial population (initial total population size = $1 \times 10^6$), composed of varying frequencies of Focal to Background strains (0.01:0.99). We then simulated bacterial dynamics for a fixed period of time ($t = 24$), which was typically sufficient for the population to stabilize, and then calculated the final frequencies of the Focal and Background species in order to determine phage utility. We first explored this model with a series of systematic parameter sweeps. Specifically, we fixed a set of baseline parameters for the model (Table 1, fixed parameters), with each baseline parameter value chosen such that its order of magnitude was roughly equivalent to a previous experimental measurement (Table 1). We then systematically varied each life history trait in turn across a fixed range, while holding all other parameters constant.

In our later analyses (Fig. 5f and 6), we simulated dynamics using measured experimental parameters. In each case, absorption rate, burst size, and degradation rate were directly measured (see below), while parameters that were challenging to measure directly such as the probability of lysis and induction rate were estimated in relative terms (see below). For example, the induction rate was estimated as small and equal for both phages, as the death rate of both the lysogen and non-lysogen was similar (Fig. 5a). Probability of lysis was estimated as 25% higher for Phi2 than Phi4, as experiments showed fewer colonies growing on agar containing Phi2 than Phi4. All simulations were performed using MatLab (R2022a).

### Statistics and visualization

All graphs were plotted in R version 4.3.0 "Already Tomorrow" using the ggplot package. The relationship between phage rank volatility and lysogen starting ratio (SR) was determined by fitting the linear model RV ~ SR. Differences in life history traits were determined via Student's $t$-test using the ggpubr package. The relationship between final population size (OD), lysogen identity (strain), and SR was determined by fitting the linear model OD ~ strain × SR. All statistics were computed using either base R or the ggpubr package.

### Strains, media, and culturing conditions

Standard cultures of lysogenized and non-lysogenized *Pseudomonas aeruginosa* PAO1 (GFP, GFP-Phi2, GFP-Phi4, and dTomato), kindly provided by C. James (5), were incubated overnight at 37°C with 180 rpm shaking in 6 mL King's medium B (KB) in 30 mL plastic microcosms. Bacterial densities were measured by plating a serial dilution of each culture onto 1.2% agar KB plates to give colony-forming units (CFU mL$^{-1}$). Phage stocks were extracted by filtering the lysogenized bacteria overnight culture through syringe filters (0.22 µm) and stored at 5°C. Phage densities were measured by plating serially diluted stocks onto a susceptible bacterial lawn embedded within soft agar (0.6% agar KB containing 1% GFP-PAO1).

## Absorption rate calculation

Absorption rate, the rate at which phages bind to bacterial cells, was calculated by mixing susceptible bacteria with phage stocks and measuring the reduction in free phage particles detected. First, $1 \times 10^4$ PFU of each phage was mixed with $1 \times 10^7$ CFU of GFP-PAO1 in 96-well plates. Mixtures were incubated at room temperature for 10 min, then filtered using a 96-well filter plate (0.22 μm filter, centrifuged at 4,500 rcf for 5 min). Phage densities were measured from starting stocks (initial PFU), and the filtrate was collected after 10-min incubation (final PFU). The absorption rate was calculated using the following calculation:

$$\text{Absorption rate} = \frac{\text{final PFU} - \text{initial PFU}}{\frac{1}{6} \times \text{initial PFU} \times \text{initial CFU}}$$

## Burst size calculation

Burst size, the amount of phages produced in an infection cycle, was determined by calculating the ratio of PFU:CFU at 0 and 3 h. Phage stocks (Phi2 or Phi4) were first mixed with GFP-PAO1 at a ratio of 1,000:1 bacteria:phage ($1 \times 10^7$ CFU and $1 \times 10^4$ PFU, respectively). The mixtures were grown under standard conditions for 3 h after which half of each culture was syringed through a 0.22 μm filter. Phage densities were measured from starting stock (initial PFU) and 3-h filtrate (final PFU). Bacterial densities were measured at the start (initial CFU) and from the remaining 3-h culture (final CFU). The burst size was calculated using the following calculation:

$$\text{Burst size} = \frac{\text{final PFU} - \text{initial PFU}}{\text{initial CFU} - \text{final CFU}}$$

## Degradation rate calculation

The degradation rate (the rate phage particles become inactive) was calculated by comparing the PFU at 0 and 24 h. Pure phage lysates were extracted by filtration (0.22 μm) from lysogenized bacterial culture grown under standard conditions. Degradation rates were calculated by incubating pure phage lysates, extracted from PAO1-Phi2 and PAO1-Phi4, without bacteria present for 24 h under standard culturing conditions for lysogenized bacteria (37℃, 180 rpm) in a 1 mL Eppendorf tube. The initial phage density ($\text{PFU}^{T0}$) and endpoint phage density ($\text{PFU}^{T24}$) were measured; the difference in PFU was transformed into a rate per hour to give degradation rate.

$$\text{Degradation rate} = \frac{\text{PFU}^{t0} - \text{PFU}^{t24}}{24}$$

## Induction rate

We estimated the induction rate by quantifying the rate of spontaneous cell lysis. To achieve this, we quantified the production of free DNA during a bacterial growth curve using Sytox DNA stain. This method has previously been used to look at the lysis rate of cells by both bacteriocins and bacteriophages (25, 26). Here, we use it as a proxy to understand if lysogens are dying significantly faster than either other lysogenized cells or non-lysogens. Individual cultures of PAO1-GFP, PAO1-Phi2, and PAO1-Phi4 (at $1 \times 10^7$ CFU mL$^{-1}$) were inoculated with 10 μM Sytox green nucleic acid stain (Thermofisher). These were left at room temperature for 15 min to enable the absorption of the stain and then grown at 37℃ in a plate reader (Clariostar) for 10 h with shaking. The emission of Sytox was recorded (excitation 490 nm, emission 537 nm) at 5 and 10 h post-inoculation. The induction rate was estimated with the following calculation:

$$\text{Induction rate} = \frac{\text{emission}^{t10} - \text{emission}^{t5}}{5}$$

## Probability of lysis calculation

The probability of lysis, our estimation of the frequency of lytic vs lysogenic infections, was calculated by plating a dilution series of PA01-wt onto soft agar lawns (0.6% agar KB) embedded with each phage ($1 \times 10^8$ PFU mL$^{-1}$) or a phage-free control (either 1% KB broth) incubated for 18 h at 37°C, static. Bacterial colonies were enumerated, and the probability of lysis was calculated through the following calculation:

$$\text{PoL} = \frac{\text{CFU}^{+\text{lysogen}}}{\text{CFU}^{-\text{lysogen}}}$$

## Competition experiments

To investigate the competitiveness of the different strains (and thereby determine phage effectiveness), we first grew our focal strains in co-culture with our background strain at a variety of different starting ratios. We then used flow cytometry to calculate the final frequency of the focal strain under each condition. Specifically, overnight cultures of lysogenized and non-lysogenized strains of bacteria were mixed at six different ratios (100:0, 1:99, 1:9, 1:1, 1:9, and 1:99). Our focal strain contained a GFP fluorescent marker, and our background strain contained a dTomato. Initial population ratios were validated by flow cytometry (see below). Mixtures were diluted 100-fold into KB media (6 mL in 30 mL glass vial) and incubated for 24 h at 37°C, 180 rpm. After 24 h, the endpoint population ratios were determined by flow cytometry. To determine growth profiles, a sample of each starting mixture was also inoculated into a 96-well plate with KB (1 in 100 dilution) and incubated in a plate reader (Tecan SPARK) for 24 h at 37°C, 180 rpm to track bacterial growth (OD$_{600}$) every 30 min.

## Flow cytometry

Chromosomal fluorescent labels (GFP or dTomato inserted within Tn7) were used to distinguish competing bacterial strains by flow cytometry and provide a population ratio. Samples were first fixed by centrifuging cultures (5,000 rpm, 3 min) and resuspending the pellet in 4% paraformaldehyde (Thermofisher). Following 30-min incubation at room temperature to complete fixation, samples were washed twice by centrifugation and resuspended in phosphate-buffered saline (PBS). Diluted samples (1 in 100 in PBS) were analyzed on a Cytoflex flow cytometer (Beckman Coulter) to count GFP-PA01 cells (ex. 488 nm, em. 525/40 nm) and dTomato-PA01 cells (ex. 488 nm, em. 610/20 nm). Cell counts were calculated using Kaluza Analysis (version 2.1) with electronic gates defined from single culture preparations of each strain. The population ratio was determined as the proportion of GFP-PAO1 cells within the total population (i.e., GFP-PAO1 + dTomato-PAO1).

## Nutrient-explicit model

Motivated by our experimental observations we adapted our model to explicitly capture growth on a defined nutrient source. This yielded the following system of differential equations capturing the dynamics of each bacterial population and the nutrient itself:

Focal strain $= \dfrac{dF}{dt} = F(rg - \kappa)$

Background strain $= \dfrac{dB}{dt} = B(rg - a\pi P - a(1 - \pi)P)$

Novel lysogen $= \dfrac{dN}{dt} = Ba(1 - \pi)P + N(rg - \kappa)$

Bacteriophage $= \dfrac{dP}{dt} = \beta\lambda\tilde{B} + \beta\lambda\tilde{F} - aPB - \delta P$

Induced/infected background strain $= \dfrac{d\tilde{B}}{dt} = \kappa N + a\pi BP - \lambda\tilde{B}$

Induced focal strain $= \dfrac{d\tilde{F}}{dt} = \kappa F - \lambda\tilde{F}$

Nutrients $= \dfrac{dR}{dt} = -0.000000005(Frg + Brg + Nrg)$

$g = \dfrac{R}{R + k}$

Here, $F$, $B$, etc., are defined as before, while $R$ represents the amount of nutrients present within the environment, and $g$ captures a saturating nutrient uptake function. In all analyses using this nutrient-explicit model, phage life history traits were as before (Table 1), while nutrient uptake and conversion parameters were selected to qualitatively match the growth of the non-lysogen PAO1 when in isolation. As previously, all simulations were performed using MatLab (R2022a).

## AUTHOR AFFILIATION

[1]Division of Evolution and Genomic Sciences, Faculty of Biology, Medicine and Health, University of Manchester, Manchester, United Kingdom

## AUTHOR ORCIDs

M. J. N. Thomas http://orcid.org/0009-0007-6579-6125
M. A. Brockhurst http://orcid.org/0000-0003-0362-820X
K. Z. Coyte http://orcid.org/0000-0002-5231-9350

## FUNDING

| Funder | Grant(s) | Author(s) |
| --- | --- | --- |
| Wellcome Trust (WT) | 226047/Z/22/Z | K. Z. Coyte |
| Wellcome Trust (WT) | 220243/Z/20/Z | M. A. Brockhurst |
| UKRI \| Biotechnology and Biological Sciences Research Council (BBSRC) | BB/T014342/1 | M. A. Brockhurst |

## DATA AVAILABILITY

All code and experimental data underpinning this work can be found at https://github.com/MJNT1999/LysogenInvasionModel.

## ADDITIONAL FILES

The following material is available online.

### Supplemental Material

**Supplemental material (mSystems01036-23-s0001.pdf).** Fig. S1 to S10.

### Open Peer Review

**PEER REVIEW HISTORY (review-history.pdf).** An accounting of the reviewer comments and feedback.

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
