## [Reviewer comments · mSystems]

What makes a temperate phage an effective bacterial weapon?

Matthew Thomas, Michael Brockhurst, and Katharine Coyte

Corresponding Author(s): Katharine Coyte, The University of Manchester

Review Timeline:

Submission Date:	October 3, 2023
Editorial Decision:	November 8, 2023
Revision Received:	February 1, 2024
Editorial Decision:	February 29, 2024
Revision Received:	March 15, 2024
Accepted:	April 2, 2024

Editor: William Harcombe

Reviewer(s): Disclosure of reviewer identity is with reference to reviewer comments included in decision letter(s). The following individuals involved in review of your submission have agreed to reveal their identity: Jorge Moura de Sousa (Reviewer #1); Sam Brown (Reviewer #2)

Transaction Report:

DOI: <https://doi.org/10.1128/msystems.01036-23>

Re: mSystems01036-23 (What makes a temperate phage an effective bacterial weapon?)

Dear Dr. Katharine Z Coyte:

There is clear interest in the findings of this manuscript, but as mentioned by each reviewer there is substantial room for improved clarity.

Revision Guidelines

Sincerely,
William Harcombe
Editor
mSystems

Reviewer #1 (Comments for the Author):

The work by Thomas and colleagues aims at understanding the interactions between the different ecological components that makes prophages efficient (or not) self-amplifying weapons for bacterial competition. I enjoyed reading their manuscript, and I really appreciated their work, particularly how the two main approaches (modelling and experiments) are used iteratively to inform one another. The results are very interesting too, shedding important light on less immediately obvious ecological

interactions that underlie phage-bacteria interactions. The manuscript is well written, the flow has logic, and the key messages are, for the most part, clear. I don't find any major issues with the manuscript, but I do wish to point some aspects that I feel might improve the clarity, interpretation and impact of the results.

Main points:

- I find the nomenclature on Figure 1 a bit confusing. It is not immediately clear what B^L and B^{nl} stand for in the formula (panel c). From the plot in panel b, I assume that B^{nl} correspond to "Novel Lysogen" (since it is the strain with the largest final abundance), but B^L is less clear. Does it refer to Background Lytic? That would explain why its final frequency is 0 in panel b, but then why does it have the same subscript of Focal lysogens (F^L)? It makes more sense to me to think of it as B wild type (Bwt) but the subscript is not the same - even if the color in plots in panel b is. I think all of this can be clarified if the authors provide a correspondence of the nomenclature in the PEI formula directly in the scheme in panel a (or at the very least in the legend of the figure, which for now only refers to the focal lysogen vs the focal wt).
- The Probability of Lysis (PoL) is identified as one of the main parameters in the model. But it was not clear to me if this is a direct correspondence to the L parameter (lines 108-109), or a compound of many parameters (e.g., induction [i], lysis [L] and/or lysis rate [p]). I think it might be the former, but in that case it would better to make it explicit, either by using a similar nomenclature in the model and in the text, or making an explicit statement of equivalence.
- The uncertainty regarding the (mathematical) meaning of PoL led me to not completely understand one of the conclusions of the paper, namely the one on lines 156-159. Assuming my interpretation is correct and that PoL is equivalent to L (meaning, the probability that infected cells will lyse instead of becoming lysogens), it is not evident why a high PoL would have a detrimental effect when the invading lysogen was present at higher abundances ("... phage with a lower PoL provided the greatest advantage when lysogens were introduced to the community at higher abundances"). If the effect of PoL resides on the recipient strain only, assuming that all invading lysogens are resistant to new infections, shouldn't a high PoL be more advantageous across all scenarios, regardless of the frequency of the invader (since the Background strain would be eliminated faster)? Maybe I am missing a key point here, and the authors could help me understand it.
- In line 197 (and subsequent ones), the authors describe "... large mitigating effects..." of certain traits on PEI when varied alongside other traits (e.g., induction rate as shown in Fig 3). Although I agree that in the examples shown there is a transition from a negative to a positive PEI, the range is quite small, typically within -0.05 to 0.05. This is particularly evident when in contrast with the effect of PoL (panel a) where the range of PEI varies from 0 to 1. The color scale is also a bit misleading on the magnitude of effects, although I understand that at the latter smaller scale (between -0.05 and 0.05) it would be hard to visually pinpoint any differences. My main point here is that the magnitude of change, even if it swings from positive to negative PEI, might hover around a certain "neutrality" of the PEI when analysing how a population is effectively impacted by these variations. Do the authors know if this impacts the population dynamics significantly, to warrant the term "large mitigating effects"? To be clear, I think the effect on PEI exists, and it is important to highlight. And even if there is no strong effect in population dynamics, it is not an issue of the analysis. But I would refrain to call these "large mitigating effects".
- One point that I would like the authors to address, even if just by discussing it, is the possibility that the prophage induction rate is not similar between the Focal and Background strain. It is mentioned in the introduction that there is "... good evidence that lysogens can be more competitive", which is not necessarily so. Recent lysogens (i.e., that are novel hosts to the lysogenic phage) tend to show higher induction rates, likely due to the fact that prophage and host have not had yet time to evolve a more "domesticated" relationship (see, e.g., 10.1038/s41467-022-33412-8) and in some cases these recent lysogens are quickly outcompeted due to their high death rate (see, e.g. 10.7554/eLife.83479 which is already cited in the manuscript). Given these known cases, were the model to consider distinct induction rates in the Focal (I^F) and Background (I^B) strains, are the results expected to remain qualitatively similar? To be clear, I am not requesting a massive reanalysis of the model, which is not trivial when consider this dissimilarity between strains. A few simulations would be nice, as it would to have at least an idea on whether similar or dissimilar induction rates are observed in the experimental strains competed here, to support this simplification in the model. But again, even just a discussion of this issue would be important, and appreciated.
- My final point is more of a curiosity, since it would be hard to do a systematic analysis of this along all the possible variations of parameters: given that PEI is assumed to be the "final" frequency of the focal strain, is there anything interesting when looking at the its temporal dynamics? At $t=0$, in all cases PEI should be 0, but then does it typically tend to be a monotonous dynamic until $t=final$, or are there cases that show more complex temporal dynamics?

Some minor points:

- Line 89/140/141/...: Probability of Lysis is capitalized in some parts of the text (e.g., lines 89/140/...) but not in others (e.g., lines 202/270/...)
- Regarding Figure 4 (which, by the way, is a really nice analysis and representation!), on the y-axis PUI should be PEI, right?
- Lines 271/276: Should be Fig 5 (and not Fig 3), right?
- Line 361: Small typo on "habour"
- Line 696: Small typo on "Predicated"

Reviewer #2 (Comments for the Author):

The authors develop population dynamical theory to predict what are the temperate phage features that make them an effective bacterial weapon, and then go on to test these predictions experimentally using a clinically relevant model system (*Pseudomonas aeruginosa* with/without temperate phages from the Liverpool Epidemic Strain). Together, these results point to the importance of lysis and induction parameters in shaping effective phage weapons (in general, high lysis, low induction). The analyses further provide important insights into the context-dependence of these results, in particular the role of strain frequency and nutrient conditions.

Overall I enjoyed reading the paper and I am sure it will be a valuable addition to the field. The rank volatility analysis was a fun and innovative way to systematically explore parameter combinations. I do have a few concerns that need to be addressed (or rebutted) in a revised manuscript.

1. General presentation.

I appreciate that the authors have worked hard to make technical details accessible to a broad audience in the main text, yet as someone who wants to follow along with the equations, this led to some frustration. I suggest that in the methods / model analysis section you provide a complete standalone table with all notation in one place. I got stuck in the first equation, as I could not find a definition for M , despite spending some time scrolling around. I assume it is a competition term, so better to use a negative sign? Conversely, there are terms in the main text such as 'probability of lysis' (abbreviated to PoL) where it would be helpful to use the actual model notation L , etc. Finally - and this is just a preference - using Y1 through Y6 doesn't help rapid processing of what each equation is describing. Either use more mnemonic variable names, or perhaps colour code with a figure? The less time we spend decoding the model, the more fun we can have reading the paper.

2. Choice of default parameters

The math results are all numerical, so it becomes super important to justify parameter values. My concern was first flagged on line 161 where 'induction rate = 0.3' was mentioned - this seems very high compared to measured literature values. Similarly in Table 1, 'probability of lysis' is defined with a reference (fixed) value of 0.5 - very high. I think this deserves some more discussion and potential revision. My sense from the literature is that measured induction rates are just above zero and measured lysis probabilities are close to 1. I think it is interesting to examine behaviors across broad parameter ranges, but using experimentally derived anchor values would help for context.

3. Frequency dependence.

I think some clarification on why the shape of frequency dependence is different from that described earlier by Brown et al. (2006, cited). The current analyses point to an intermediate frequency maximum via the PEI metric (which I think is useful), while the earlier study pointed to faster lysogen invasion from rare, and greater weapon production per lysogen invader when rare. I think this is a case of different metrics highlighting different properties, which is always worth spelling out clearly. I think the data from Brown (2006) is sufficient to measure PEI from experimental data (there was phage KO controls), which could be helpful to parse whether the differences are due to different metrics or different biology / parameters.

4. Role of nutrients.

I appreciate the attentiveness to novel experimental results and pursuit of additional math model insights. But I would note that earlier work did implicitly capture nutrient variation, by varying carrying capacity. See Fig 1 in Brown (2006) for analysis of frequency and density dependence.

5. Phage agency.

The authors allude to phage strategies / agency around line 387, which is good to see. I'd encourage a bit more on this avenue as I think it is a very important consideration to build a complete picture of the biology. By viewing prophages purely as a bacterial 'tool' we might systematically miss key forces shaping prophage life-history evolution. This crossed my mind around lines 360 - I think it is plausible that prophages do persist by playing a sophisticated 'conditional parasite' strategy. See in particular existing work on 'why be temperate', eg Li et al (2020) DOI 10.1093/ve/veaa042 which offers a nice math treatment of phage life history parameter evolution from a phage perspective.

For a general perspective that isn't so often brought up in this context, see Van Baalen & Jansen (2003) DOI 10.1034/j.1600-0706.2001.950203.x. This co-evolutionary view offers some paths to think about prophage cargo strategies (toxins etc), and the role of bacteria/phage co-adaptation in limiting the proliferation of novel lysogens (see Brown et al. 2009 DOI 10.1111/j.1752-4571.2008.00059.x for some discussion on this).

I'll sign as I have failed to avoid multiple self citations in my review - Sam Brown

We thank both reviewers for their positive assessments of our manuscript, we're very glad they both enjoyed reading it, and found each of their comments very constructive. We have addressed each comment in our point-by-point below – we believe doing so has substantially improved the clarity of our manuscript, and added some interesting new angles to the work.

Reviewer #1 (Comments for the Author):

The work by Thomas and colleagues aims at understanding the interactions between the different ecological components that makes prophages efficient (or not) self-amplifying weapons for bacterial competition. I enjoyed reading their manuscript, and I really appreciated their work, particularly how the two main approaches (modelling and experiments) are used iteratively to inform one another. The results are very interesting too, shedding important light on less immediately obvious ecological interactions that underlie phage-bacteria interactions. The manuscript is well written, the flow has logic, and the key messages are, for the most part, clear. I don't find any major issues with the manuscript, but I do wish to point some aspects that I feel might improve the clarity, interpretation and impact of the results.

Thank you again for your positive comments

Main points:

- I find the nomenclature on Figure 1 a bit confusing. It is not immediately clear what B^L and B^{NL} stand for in the formula (panel c). From the plot in panel b, I assume that B^{NL} correspond to "Novel Lysogen" (since it is the strain with the largest final abundance), but B^L is less clear. Does it refer to Background Lytic? That would explain why its final frequency is 0 in panel b, but then why does it have the same subscript of Focal lysogens (F^L)? It makes more sense to me to think of it as B wild type (Bwt) but the subscript is not the same - even if the color in plots in panel b is. I think all of this can be clarified if the authors provide a correspondence of the nomenclature in the PEI formula directly in the scheme in panel a (or at the very least in the legend of the figure, which for now only refers to the focal lysogen vs the focal wt).

Our apologies for the confusion here, as outlined in further detail below and in our responses to Reviewer 2, we realise our notation in both Figure 1 and the main text was a little unclear.

Regarding the specific question above, B^L referred to the (phage free) Background Strain in the case where the focal strain was a lysogen. More broadly, to increase clarity we have now edited our nomenclature in both Figure 1 (see below) and the equations themselves, which we have moved from the materials and methods into the main text.

In our new version, throughout F always corresponds to the Focal strain, B to the background strain, and N to the novel lysogen formed when the phage integrates into the background strain. Superscripts in Figure 1 indicate whether we are considering the case where the Focal strain is a lysogen (superscript L) or wildtype (superscript WT). We have also rephrased our subpanel titles to clarify the different test cases.

Fig 1. Mathematical modelling captures the dynamics of phage-mediated competition. **a.** Schematic illustrating our mathematical model. Here we focus on the dynamics of two competing strains, an initially susceptible “Background” strain (dark red), and a “Focal” strain (dark blue) which may or may not carry a temperate phage weapon. **b.** Example bacterial population dynamics from simulations when the Focal strain does (right - L) or does not (left - Wt) carry a temperate phage weapon. Dashed lines indicate final population abundances, which are in turn used to calculate the Phage Effectivity Index (PEI). **c.** Formula for the PEI, defined as the difference in the final frequency of the Focal strain with vs without a temperate phage. Here F indicates the final abundance of the focal strain, B the background strain, and N the final abundance of the novel lysogen formed when the phage integrates into the background strain. Superscripts indicate simulations in which the Focal strain does (L) or does not (WT) harbour a temperate phage.

- The Probability of Lysis (PoL) is identified as one of the main parameters in the model. But it was not clear to me if this is a direct correspondence to the L parameter (lines 108-109), or a compound of many parameters (e.g., induction [i], lysis [L] and/or lysis rate (p)). I think it might be the former, but in that case it would better to make it explicit, either by using a similar nomenclature in the model and in the text, or making an explicit statement of equivalence.

You are correct in that PoL referred to a single parameter, the lysis probability L (please note, to avoid reuse of symbols in our latest version we now use the character π), and is not a compound of many parameters. Again, our notation was, in hindsight, not particularly clear.

As suggested, we have now removed the use of the PoL abbreviation throughout and instead refer directly to the relevant model parameters wherever appropriate in the text (e.g. “we found that small changes in the **probability of lysis (π)** and the induction rate (**κ**) both led to large changes in the effect of a given phage”). To further clarify, we have also added an additional column to Table 1 to indicate the meaning of each parameter and the symbol used to represent it.

- The uncertainty regarding the (mathematical) meaning of PoL led me to not completely understand one of the conclusions of the paper, namely the one on lines 156-159. Assuming my interpretation is correct and that PoL is equivalent to L (meaning, the probability that infected cells will lyse instead of becoming lysogens), it is not evident why a high PoL would have a detrimental effect when the invading lysogen was present at higher abundances (“... phage with a lower PoL provided the greatest advantage when lysogens were introduced to the community at higher abundances”). If the effect of PoL resides on the recipient strain only, assuming that all invading lysogens are resistant to new infections, shouldn't a high PoL be more advantageous across all scenarios, regardless of the frequency of the invader (since the Background strain would be eliminated faster)? Maybe I am missing a key point here, and the authors could help me understand it.

Your intuition is entirely correct, increasing the probability of lysis (π) never decreases the PEI – that is, a higher probability of lysis is never detrimental. What we intended to convey with the sentence “...phage with a lower PoL...” is that when π is high, possessing a phage confers the greatest benefit to the focal lysogen when the lysogen is initially rare, whereas when π is low, the benefits are greatest when the focal lysogen is present at intermediate frequencies.

We have now substantially rewritten this section, using the example of induction rate rather than probability of lysis, as we focus more on this parameter in subsequent sections (Lines 181 – 187):

Importantly, however, regardless of PEI metric used, we found that the relationship between individual phage life history traits and phage weapon effectivity was strongly modulated by initial lysogen frequency. For example, while harbouring a phage with a high induction rate was always detrimental, this disadvantage was much lower when lysogens were common within the population than when they

were rare. This interplay between the characteristics of a lysogen and its environment led to situations whereby a given phage could either be costly or beneficial, depending on the starting frequency of the lysogen within the population.

- In line 197 (and subsequent ones), the authors describe "... large mitigating effects..." of certain traits on PEI when varied alongside other traits (e.g., induction rate as shown in Fig 3). Although I agree that in the examples shown there is a transition from a negative to a positive PEI, the range is quite small, typically within -0.05 to 0.05. This is particularly evident when in contrast with the effect of PoL (panel a) where the range of PEI varies from 0 to 1. The color scale is also a bit misleading on the magnitude of effects, although I understand that at the latter smaller scale (between -0.05 and 0.05) it would be hard to visually pinpoint any differences. My main point here is that the magnitude of change, even if it swings from positive to negative PEI, might hover around a certain "neutrality" of the PEI when analysing how a population is effectively impacted by these variations. Do the authors know if this impacts the population dynamics significantly, to warrant the term "large mitigating effects"? To be clear, I think the effect on PEI exists, and it is important to highlight. And even if there is no strong effect in population dynamics, it is not an issue of the analysis. But I would refrain to call these "large mitigating effects".

*This is a very fair point and one that we have wondered about ourselves. While our intuition is that the critical difference between being slightly beneficial vs slightly detrimental will have a substantial impact on longer term eco-evolutionary dynamics, to say so concretely requires much a larger body of analysis beyond the scope of this work. For the time being we have edited our text to remove the phrase "large mitigating effects" (e.g. Lines 218 – 221) **More surprisingly, traits such as absorption rate and lysis rate, which had very little effect on PEI when varied in isolation, also had large mitigating effects on PEI could mitigate negative effects of high induction rate (Fig 3b,c). For example, small increases in the absorption rate switched PEI of high induction rate phages from being negative to positive***

We have also added a short section to our discussion highlighting this point and speculating on the importance of the size of these effects (Lines 407 – 416):

...and the relative ranking of different phage weapons switches with different starting conditions. It is worth noting that in our simulations the variation in phage weapon effectiveness between different phages, and between different conditions, is relatively small. Though even subtle differences in an organism's fitness could have substantial impacts upon its long-term dynamics, to fully understand the causes of consequences of phage weapon diversity the next step will be to directly compete lysogens with different strategies against one another within an evolutionary modelling framework. Nonetheless, our work suggests these context dependent determinants of a phage weapon's effectiveness may be an important contributor to the large diversity in temperate phage life history traits observed in nature.

- One point that I would like the authors to address, even if just by discussing it, is the possibility that the prophage induction rate is not similar between the Focal and Background strain. It is mentioned in the introduction that there is "... good evidence that lysogens can be more competitive", which is not necessarily so. Recent lysogens (i.e., that are novel hosts to

the lysogenic phage) tend to show higher induction rates, likely due to the fact that prophage and host have not had yet time to evolve a more "domesticated" relationship (see, e.g., 10.1038/s41467-022-33412-8) and in some cases these recent lysogens are quickly outcompeted due to their high death rate (see, e.g. 10.7554/eLife.83479 which is already cited in the manuscript). Given these known cases, were the model to consider distinct induction rates in the Focal (I^{Focal}) and Background ($I^{\text{Background}}$) strains, are the results expected to remain qualitatively similar? To be clear, I am not requesting a massive reanalysis of the model, which is not trivial when consider this dissimilarity between strains. A few simulations would be nice, as it would to have at least an idea on whether similar or dissimilar induction rates are observed in the experimental strains competed here, to support this simplification in the model. But again, even just a discussion of this issue would be important, and appreciated.

This is a very good point, as it is known that novel lysogens are likely to have markedly higher induction rates than existing lysogens. To explore this we have conducted the additional analyses suggested, repeating our analyses of the interplay between key life history traits, initial lysogen frequency and PEI under the scenario where the novel lysogen has a slightly or a substantially higher induction rate (see Figure below).

Though in no way exhaustive, our simulations suggest that asymmetric induction rates do not change the relationship between probability of lysis and PEI – that is a higher probability of lysis is always beneficial, and the utility of a given phage weapon appears to be greatest when lysogens are initially present at intermediary frequencies within a population.

The impact of asymmetric induction rates on the interplay between induction rate and PEI is slightly more interesting. Specifically, we find that if induction rates in the novel lysogen are much higher than those of the focal strain, then this substantially increases the utility of a given phage weapon, to the point that it can compensate for the otherwise detrimental effects of high induction rates. That is, while it remains the case that low induction rate phage tend to be more effective, even high induction rate phages are useful (PEI always > 0). This result is consistent with our existing analyses of induction rate compensation, in that again we see lysogens can bear the cost of a high induction rate phage, provided they are able to rapidly infect and kill competitors.

We have included this additional analysis as SI figure 4, and reference this result in our section on life history trait compensation (Lines 221 – 224), "***Notably, the detrimental effect of high induction rates in the focal strain could also be overcome when the novel lysogens possessed even higher induction rates still, a phenomenon that has been observed several times in nature (Fig SI 4).***"

SI 4. Phage weapons are more useful when novel competitor lysogens have a higher induction rate than Focal strain lysogens. Newly generated lysogens are observed to have higher induction rates than more established lineages, prompting us to investigate the effect of induction rate asymmetry on PEI. In all tested conditions a higher induction rate in the novel lysogen amplified the benefit of the phage. This amplified benefit can compensate for the otherwise detrimental effect of high induction rates in the focal strain, such that even phage with very high induction rates can be beneficial. Rows correspond to the relationship between probability of lysis (a) or induction rate (b) and PEI in three different conditions. The number above each plot shows the factor the induction rate of the novel lysogen is increased by (Novel lysogen induction rate = Focal induction Rate + Number).

- My final point is more of a curiosity, since it would be hard to do a systematic analysis of this along all the possible variations of parameters: given that PEI is assumed to be the "final" frequency of the focal strain, is there anything interesting when looking at its temporal dynamics? At $t=0$, in all cases PEI should be 0, but then does it typically tend to be a monotonous dynamic until $t=final$, or are there cases that show more complex temporal dynamics?

Thanks for this very interesting suggestion – you're correct that doing this systematically is tricky, but to explore this somewhat we performed a small analysis plotting PEI over time under a small selection of different parameter combinations, as illustrated in the figure below:

In most cases we found that the effectivity of a phage weapon increases monotonically over time, rising rapidly at the start of the competition and plateauing towards the end, reflecting the course of phage infection within the susceptible population.

However, we found some particularly interesting behaviour for phages with intermediate to high induction rates. In these cases, the phage weapon was highly detrimental early in the competition (i.e. very negative PEI) but this cost gradually reduced over time, likely reflecting how the initial cost of the phage weapon (in lysing large numbers of host cells) is ameliorated as free virions begin to infect and lyse susceptible competitors. In some cases, this non-linear behavior of PEI led to scenarios where the phage weapon could appear detrimental or beneficial depending upon when PEI was measured. Moreover, it could also generate situations where the optimal phage weapon briefly switched over time. While we're hesitant to draw any major conclusions from this relatively small-scale analysis, we do think this is an interesting result so have we now included this analysis as supplementary figure 9, and nod towards it in our discussion section (Lines 450-453):

*To explore systematically the determinants of phage effectiveness here we have used relatively simple ecological models ... **We have also focused on very simple snapshot measures of phage effectivity, calculating PEI only at the end of individual competitions. While beyond the scope of this study, preliminary analysis suggests PEI may also vary over time in interesting ways (Fig SX).** Understanding how these many added layers of complexity further modulate the effectiveness of phage weapons is therefore an important open question.*

SI 9. PEI varies over time, but relationships between PEI and individual life history traits remain consistent. a-c Changes in PEI over time for three different phage, varying in their probability of lysis (π , colors) under three different starting ratios. In these cases PEI monotonically increases over time as the phage spreads within the system, and the higher the probability of lysis, the higher the PEI. **d-f** Changes in PEI over time for three different phage, varying in their induction rate (κ , colors) under three different starting ratios. When induction rates are high the phage is initially highly detrimental, however this cost lessens over time as the phage spreads within the system and passes on the cost of the phage to the novel lysogens. Notably this can result in the phage being costly or beneficial depending on how long the competition is observed, and can even lead to scenarios where the optimal phage weapon switches over time.

Some minor points:

- Line 89/140/141/...: Probability of Lysis is capitalized in some parts of the text (e.g., lines 89/140/...) but not in others (e.g., lines 202/270/...)

Capitalization is now consistent throughout

- Regarding Figure 4 (which, by the way, is a really nice analysis and representation!), on the y-axis PUI should be PEI, right?

Thanks and yes, we have edited the y-axis to read PEI

- Lines 271/276: Should be Fig 5 (and not Fig 3), right?

Yes, we have now fixed this reference

- Line 361: Small typo on "habour"

We have corrected this spelling

- Line 696: Small typo on "Predicated"

Thanks, we've now fixed this.

Jorge Moura de Sousa

Reviewer #2 (Comments for the Author):

The authors develop population dynamical theory to predict what are the temperate phage features that make them an effective bacterial weapon, and then go on to test these predictions experimentally using a clinically relevant model system (*Pseudomonas aeruginosa* with/without temperate phages from the Liverpool Epidemic Strain). Together, these results point to the importance of lysis and induction parameters in shaping effective phage weapons (in general, high lysis, low induction). The analyses further provide important insights into the context-dependence of these results, in particular the role of strain frequency and nutrient conditions.

Overall I enjoyed reading the paper and I am sure it will be a valuable addition to the field. The rank volatility analysis was a fun and innovative way to systematically explore parameter combinations. I do have a few concerns that need to be addressed (or rebutted) in a revised manuscript.

Thank you again for your positive comments

1. General presentation.

I appreciate that the authors have worked hard to make technical details accessible to a broad audience in the main text, yet as someone who wants to follow along with the equations, this led to some frustration. I suggest that in the methods / model analysis section you provide a complete standalone table with all notation in one place. I got stuck in the first equation, as I could not find a definition for M , despite spending some time scrolling around. I assume it is a competition term, so better to use a negative sign?

Conversely, there are terms in the main text such as 'probability of lysis' (abbreviated to PoL) where it would be helpful to use the actual model notation L , etc. Finally - and this is just a preference - using Y_1 through Y_6 doesn't help rapid processing of what each equation is

describing. Either use more mnemonic variable names, or perhaps colour code with a figure? The less time we spend decoding the model, the more fun we can have reading the paper.

Thank you for your very reasonable suggestions which are also in line with those of Reviewer 1 – we realise our original nomenclature was particularly unclear.

Regarding your specific question about M , yes this is indeed the community interaction matrix. While you're right that because we assume all bacterial strains are competing we could use $-M$, we prefer to keep the $+M$ notation (with all entries in M equal and negative) for consistency with classic gLV notation.

More generally, in line with your suggestions, to increase clarity we've made the following edits:

- We have revamped our mathematical notation so that variables are both more intuitive, and more consistent with our main text and figures. For example, rather than Y_1, Y_2 etc we now use F, B, N etc, illustrating the focal, background and novel lysogen strains respectively.
- We have moved the equations themselves into the results section (from the methods) and provide brief labels for each compartment
- We have swapped some of our parameter symbols to remove instances of duplication of characters (e.g. previously L indicated both lysogen and probability of lysis)
- We have edited Table 1 so that for each parameter symbol we have a clear glossary of its biological meaning, its fixed value, and the range across which it is varied
- We have removed instances of the PoL abbreviation and instead refer directly to the appropriate parameter (previously L , now π to avoid symbol duplication)
- We have edited the main text and Table 1 such that M (now μ) is defined.

2. Choice of default parameters

The math results are all numerical, so it becomes super important to justify parameter values. My concern was first flagged on line 161 where 'induction rate = 0.3' was mentioned - this seems very high compared to measured literature values. Similarly in Table 1, 'probability of lysis' is defined with a reference (fixed) value of 0.5 - very high. I think this deserves some more discussion and potential revision. My sense from the literature is that measured induction rates are just above zero and measured lysis probabilities are close to 1. I think it is interesting to examine behaviors across broad parameter ranges, but using experimentally derived anchor values would help for context.

This is a very fair point, we had selected our fixed points for each parameter to sit roughly midway between each of the ranges explored. However, this does mean our fixed induction rates are a little high, and our probability of lysis a little low compared to those observed in nature. The impact of varying these fixed values is partially covered by our analysis in Fig 3, where we examine how changing various life history trait modulates the detrimental effect of high induction rates. However, we agree it is also worth including a more explicit analysis with some experimentally derived anchor values.

We have now repeated our induction rate parameter sweep analysis (ie Fig 2b, focusing on this as it is our most interesting result) using fixed values for the probability of lysis taken directly from our later experimental work (0.72, 0.91). As illustrated in the figures below, consistent with our Fig 3 analysis, as we increase the probability of lysis we see the cost of a high induction rate is gradually ameliorated.

We now include this analysis as supplementary figure 6, and highlight it in our results section (Lines 293-297):

“We then used these life-history data to qualitatively parameterise our model (Materials and Methods), and then predict how the effectivity of each phage weapon varies under a set of different competition conditions. For completeness, we also used these experimental parameters to repeat our earlier life history trait parameter sweep analysis (Fig SI 6).”

SI 6. Using experimentally determined probability of lysis values does not alter the relationship between induction rate, initial frequency, and PEI. Our initial parameter sweeps used a relatively low value for probability of lysis compared with those observed in our experiments, thus here we repeat our Figure 2B analyses using these higher probability of lysis values.

3. Frequency dependence.

I think some clarification on why the shape of frequency dependence is different from that described earlier by Brown et al. (2006, cited). The current analyses point to an intermediate freq maximum via the PEI metric (which I think is useful), while the earlier study pointed to faster lysogen invasion from rare, and greater weapon production per lysogen invader when rare. I think this is a case of different metrics highlighting different properties, which is always worth spelling out clearly. I think the data from Brown (2006) is sufficient to measure PEI from

experimental data (there was phage KO controls), which could be helpful to parse whether the differences are due to different metrics or different biology / parameters.

This is a good point, we spent quite some time deciding what is the most relevant measure of phage effectivity as, as you say, different functions will inherently highlight different phenomena. We could not find the raw data for the Brown et al paper, so instead have performed two complementary analyses: first, we repeated our Fig 2 analysis calculating PEI based on the natural log of the fold change in focal strain frequency (SI Fig 3) – ie giving an approximate growth rate / speed of invasion. Second, for a subset of parameters we calculated the weapon production per lysogen invader (i.e. Rebuttal Fig 1, providing a direct comparison to Fig 2C in Brown et al 2006).

As you might expect (see figures below), consistent with your previous results we find that indeed in most (although interestingly not all) cases, when using invasion speed as a metric of PEI, PEI monotonically decreases with increasing starting frequency. Similarly, we also found that weapon production efficiency was highest at low frequencies. Together, these results suggest that the differences between our conclusions and those of your 2006 study are largely due to the different choice of metrics, rather than the underlying biology/parameters.

Given the intrinsic frequency dependence of lots of these metrics, as a follow up we had already also assessed PEI using a third metric, the relative fitness of the focal strain compared to the background strain under each scenario (SI Fig 1). Specifically, following the work of Ross-Gillespie et al (DOI: 10.1086/519860), we calculated the relative fitness as the ratio of the change in frequency of the focal strain to the change in frequency of the background strain, given by the equation,

$$v = \frac{F_2 * (1 - F_1)}{F_1 * (1 - F_2)}$$

Where F1 and F2 represent the starting and final frequencies of the focal strain respectively.

We believe this metric should enable us to more thoroughly disentangle whether differences are due to differences in biology/ parameters or metrics. Here again we find that in most cases there is a PEI maximum at intermediate frequencies, although the magnitude of these differences is much smaller (thus to aid in visualization we also provide a figure plotting the derivative of PEI against frequency, SI Fig 2).

Altogether we believe this indicates that while the exact shape of the frequency dependence varies depending upon metric used, our general conclusion – that phage efficacy depends not only upon life history traits but also ecological context – is highly robust.

We have added the following sections to our main text to more clearly spell out this important point. We have also included our alternative fitness metric figures as supplementary figures 1, 2 and 3.

Lines 127-140:

...Comparing the outcome of these competitions when the Focal strain was lysogenic to when it was non-lysogenic thus enabled us to quantify the relative advantage (or disadvantage) to a bacterium of possessing a given temperate phage, which we term the Phage Effectivity Index (PEI, Fig 1C).

We focused on this measure as it allowed us to directly capture the absolute advantage to a focal strain of carrying a phage weapon (compared to a phage free non-lysogen) regardless of context. This metric has an intrinsic frequency dependence: because there is an upper limit on frequency within the population, strains which are initially common can only increase in frequency a small amount, regardless of the effectivity of their phage weapon. Therefore, for completeness we also assessed phage effectiveness under two further metrics. First, we calculated PEI as the relative fitness (v , Fig SI 1-2) of the lysogen compared to the background strain during each competition. Second, we calculated PEI as the natural log of the focal strain's fold change in abundance within the population, giving an estimate of the average growth rate of the focal strain during each competition (Fig SI 3).

Lines 169-187:

...In contrast, when lysogens were initially present at either very low or very high frequencies we observed very little impact of harbouring a phage upon the success of a lysogen within a population ($PEI \approx 0$). The exact shape of this frequency dependence varied somewhat depending on the measure of phage effectivity used. For example, lysogen relative fitness still peaked at intermediate lysogen frequencies but reduced the magnitude of this effect (Fig SI 1 and 2). In contrast, the fold change in focal strain frequency often suggested that phage weapons are most effective when the focal strains are rare (Fig SI 3), consistent with previous observations from Brown and colleagues and likely reflecting that fold changes are over-emphasized at small values.

Importantly, however, regardless of PEI metric used, we found that the relationship between individual phage life history traits and phage weapon effectivity was strongly modulated by initial lysogen frequency. For example, while harbouring a phage with a high induction rate was always detrimental, this disadvantage was much lower when lysogens were common within the population than when they were rare. This interplay between the characteristics of a lysogen and its environment led to situations whereby a given phage could either be costly or beneficial, depending on the starting frequency of the lysogen within the population.

SI 3. Analysing the impact of life history traits and frequency when phage effectiveness is calculated as the invasion speed of the focal strain. Each heatmap is the same simulations from Figure 2 but the colour now shows the PEI calculated as the natural log of the final frequency of the Focal lysogen (F) divided by the starting frequency. As in our original analyses, we find that probability of lysis and induction rate have the greatest impact on PEI, that PEI varies with lysogen starting frequency, and that the interplay between LHT and frequency leads to cases where a phage is beneficial under some conditions but detrimental under others. However, under this measure PEI is often (although not always) greatest when lysogens are rare, as this measure of competitiveness highlights relatively small changes by initially rare species.

Rebuttal Fig 1. Phage production per lysogen is fastest when lysogens are initially rare. Here the y-axis denotes the log ratio of phage to focal strain abundance, which is analogous to the $\log(V/C)$ metric present in Brown et al (2006). We show these dynamics at 3 different probabilities of lysis (columns). Our model also shows that phage amplification is fastest when the initial frequency of lysogens is low, concurrent with the findings of previous mathematical studies. We also find that this rate is increased with higher probability of lysis.

SI 1. Analysing the impact of life history traits and frequency when phage effectiveness is calculated as the relative fitness measure, v , of the lysogen. Here each heatmap is coloured by the relative fitness of the focal strain compared to the background strain, given by the equation:

$$v = \frac{F_2 * (1 - F_1)}{F_1 * (1 - F_2)}$$

Where F_1 and F_2 represent the starting and final frequencies of the Focal strain respectively. As in our original analyses, we find that probability of lysis and induction rate have the greatest impact on PEI, that PEI varies with lysogen starting frequency, and that the interplay between LHT and frequency leads to cases where a phage is beneficial under some conditions but detrimental under others. However, as relative fitness tends to saturate quickly these effects are less prominent than with other measures.

SI 2. Confirming the frequency dependence of PEI when using relative fitness. When calculating PEI as the relative fitness of the focal strain compared to the background strain, changes with frequency are small and thus challenging to detect via eye. To confirm the dependence of PEI on frequency we therefore also plot the derivative of PEI against starting frequency (i.e. $\Delta v = dPEI/dFrequency$). Here $\Delta v > 0$ indicates that PEI increases with increasing starting frequency, while $\Delta v < 0$ indicates that PEI decreases with increasing starting frequency. For all life history trait analyses, we see that the focal strains relative frequency peaks at intermediate starting frequencies.

4. Role of nutrients.

I appreciate the attentiveness to novel experimental results and pursuit of additional math model insights. But I would note that earlier work did implicitly capture nutrient variation, by varying carrying capacity. See Fig 1 in Brown (2006) for analysis of frequency and density dependence.

While we agree Brown *et al* does implicitly capture nutrient variation through varying carrying capacity, the key difference in our analysis is that we now allow nutrient availability to vary dynamically across the course of the simulation. It is this variability, and in particular the fact that nutrients can run out during a competition, that appears to drive the interesting frequency dependent abundance dynamics that we see.

That said, we by no means meant to claim we were the only people to consider nutrients as a factor, so have edited our introduction to soften this claim, and edited our discussion to clarify our key finding:

*“...these experiments reveal that an additional, ~~largely~~ **often** overlooked environmental factor – nutrient availability...”*

*“An unexpected finding was the key role of nutrients – **and more specifically, dynamic changes in nutrient availability over time** – in modulating the interplay between...”*

5. Phage agency.

The authors allude to phage strategies / agency around line 387, which is good to see. I'd encourage a bit more on this avenue as I think it is a very important consideration to build a complete picture of the biology. By viewing prophages purely as a bacterial 'tool' we might systematically miss key forces shaping prophage life-history evolution. This crossed my mind around lines 360 - I think it is plausible that prophages do persist by playing a sophisticated 'conditional parasite' strategy. See in particular existing work on 'why be temperate', eg Li et al (2020) DOI 10.1093/ve/veaa042 which offers a nice math treatment of phage life history parameter evolution from a phage perspective.

For a general perspective that isn't so often brought up in this context, see Van Baalen & Jansen (2003) DOI 10.1034/j.1600-0706.2001.950203.x. This co-evolutionary view offers some paths to think about prophage cargo strategies (toxins etc), and the role of bacteria/phage co-adaptation in limiting the proliferation of novel lysogens (see Brown et al. 2009 DOI 10.1111/j.1752-4571.2008.00059.x for some discussion on this).

Thanks for this interesting question which spurred us to revisit our original simulations to explore the dynamics of the bacteriophage themselves under each scenario. In the figure below (now SI fig 10) we plot how the total number of phage virions produced (as a measure

of phage fitness) varies with varying lysogen initial frequency, probability of lysis, and induction rate.

SI 10. The total amount of phage produced during a competition is dependent on the starting frequency of lysogens and life history traits. A. Plotting how total phage virions produced (calculated as the integral of phage abundance over time) changes with depending upon initial lysogen frequency and probability of lysis. The higher the probability of lysis, the more phage virions produced. B. Plotting how total phage virions produced changes with depending upon initial lysogen frequency and induction rate. Here phage production peaks at intermediate induction rates.

Strikingly, we find that under this admittedly relatively limited analysis, increasing induction rate has a non-monotonic relationship with virion production – with virion production peaking at an intermediate induction rate. This is in contrast to our original PEI analysis, where the phage is more useful to the lysogen when its induction rates are lower.

This benefit to the phage of intermediate induction rates is particularly interesting, as it suggests that there may be a conflict of interest between phage and bacterial host. That said, we are hesitant to make any sweeping statements about phage agency from this analysis alone given it is so very simple. A more thorough treatment would be to place our model within a co-evolutionary framework and explore how both phage and bacteria evolve over time. This is certainly something we intend to follow up with, but is beyond the scope of this paper.

As a nod to this important point, we've now expanded our discussion of phage agency, citing the paper by Li and highlighting our supplementary figure as a nod to potential interesting avenues here, while taking care not to over claim (Lines 431-443):

“Here we have focused how the characteristics of a given temperate phage impact the fitness of its bacterial host. However, an equally important question is how do these life history traits impact the spread and abundance of the phage itself? While a formal analysis of optimal phage strategies is beyond the scope of this paper, we can infer a component of phage fitness by simply tracking the number of free phage

virions produced across each of our simulations, which suggests phage fitness is similarly dependent upon ecological context. Notably, whereas bacteria benefit from very low induction rates, phage fitness, at least in some cases, is maximized at higher induction rates. This divergence is particularly interesting as it suggests the potential for conflicts of interest between temperate phages and their bacterial hosts over what is the optimal life-history. To fully understand how conflicting phage and bacterial fitness interests shape evolutionary trajectories will require a more formal coevolutionary analysis, however, our results are suggestive that such conflicts may be yet another important factor driving the phage diversity observed in nature (Li et al., 2020)."

I'll sign as I have failed to avoid multiple self citations in my review - Sam Brown

Re: mSystems01036-23R1 (What makes a temperate phage an effective bacterial weapon?)

Dear Dr. Katharine Z Coyte:

It appears that there is an issue with figure 6, and a couple of additional issues that the authors might like to fix.

Revision Guidelines

Sincerely,
William Harcombe
Editor
mSystems

Reviewer #1 (Comments for the Author):

The revised version of the manuscript satisfactorily addresses all the points I had raised in my first review. I find the nomenclature clearer now, and the additional analyses increase both the robustness and the depth of the results.

There seems to be, however, an important issue with Fig6. At least in the version I had access to, the figure doesn't match what is referenced in the text (it only has the 3 last panels, g, h and i). I don't know what happened here, but I had to analyse the

figure from the previous version of the manuscript. Which I guess it is ok for revision purposes, since I don't think this figure changed in revision, but do take notice for final submission.

Two other very minor issues:

- Minor typo in figure 1 (absorbtion -> absorption)
- Specify what the acronym LHT means, the first time it is used (in supplementary figures). I had to search a bit in the paragraph/figure legend to map it to "Life History Traits".

Jorge Moura de Sousa

Reviewer #2 (Comments for the Author):

Thanks to the authors for their compelling responses to all issues raised previously. I have no further concerns - congrats on a very nice paper.

We would once again like to thank the reviewers for looking at our manuscript and for their very constructive feedback. We are very happy that both reviewers felt our revisions satisfactorily addressed all of their comments. There were just a few final typos and formatting errors picked up by Reviewer 1, which we have now addressed, as outlined below:

Reviewer #1 (Comments for the Author):

The revised version of the manuscript satisfactorily addresses all the points I had raised in my first review. I find the nomenclature clearer now, and the additional analyses increase both the robustness and the depth of the results.

There seems to be, however, an important issue with Fig6. At least in the version I had access to, the figure doesn't match what is referenced in the text (it only has the 3 last panels, g, h and i). I don't know what happened here, but I had to analyse the figure from the previous version of the manuscript. Which I guess it is ok for revision purposes, since I don't think this figure changed in revision, but do take notice for final submission.

Thanks for catching this, we have now changed figure 6 back to the correct version (which is the same version as was included in the original submission). When revising the manuscript we had mistakenly included an older version of the figure.

Two other very minor issues:

- Minor typo in figure 1 (absorbtion -> absorption)

This typo has been fixed

- Specify what the acronym LHT means, the first time it is used (in supplementary figures). I had to search a bit in the paragraph/figure legend to map it to "Life History Traits".

We have now edited the SI figure legends to ensure the acronym LHT (Life History Traits) is specified within each figure caption where it is used.

Reviewer #2 (Comments for the Author):

Thanks to the authors for their compelling responses to all issues raised previously. I have no further concerns - congrats on a very nice paper.

We are delighted to hear Prof Brown was satisfied by our revisions and thank them for their compliments

Re: mSystems01036-23R2 (What makes a temperate phage an effective bacterial weapon?)

Dear Dr. Katharine Z Coyte:

Your manuscript has been accepted, and I am forwarding it to the ASM production staff for publication. Your paper will first be checked to make sure all elements meet the technical requirements. ASM staff will contact you if anything needs to be revised before copyediting and production can begin. Otherwise, you will be notified when your proofs are ready to be viewed.

Cover Image Submissions: If you would like to submit a potential Cover Image, please email a file and a short legend to msystems@asmusa.org. Please note that we can only consider images that (i) the authors created or own and (ii) have not been previously published. By submitting, you agree that the image can be used under the same terms as the published article. Image File requirements: TIF/EPS, 7.5 inches wide by 8.25 inches tall (at least 2,250 pixels wide by 2,475 pixels tall), minimum 300 dpi resolution (600 dpi preferred), RGB, and no figure elements, e.g., arrows or panel labels. The legend should be a short description of the image, 1-2 sentences recommended.

Sincerely,
William Harcombe
Editor